# Validation of a Visual-Based Analytics Tool for Outcome Prediction in Polytrauma Patients (WATSON Trauma Pathway Explorer) and Comparison with the Predictive Values of TRISS

**DOI:** 10.3390/jcm10102115

**Published:** 2021-05-14

**Authors:** Cédric Niggli, Hans-Christoph Pape, Philipp Niggli, Ladislav Mica

**Affiliations:** 1Department of Trauma Surgery, University Hospital Zurich, 8091 Zurich, Switzerland; cedric.niggli@swissonline.ch (C.N.); hans-christoph.pape@usz.ch (H.-C.P.); 2Department of Mathematics, ETH Zurich, 8092 Zurich, Switzerland; philipp.niggli@swissonline.ch

**Keywords:** polytrauma, WATSON Trauma Pathway Explorer, outcome, SIRS, sepsis, early death, TRISS, artificial intelligence

## Abstract

**Introduction:** Big data-based artificial intelligence (AI) has become increasingly important in medicine and may be helpful in the future to predict diseases and outcomes. For severely injured patients, a new analytics tool has recently been developed (WATSON Trauma Pathway Explorer) to assess individual risk profiles early after trauma. We performed a validation of this tool and a comparison with the Trauma and Injury Severity Score (TRISS), an established trauma survival estimation score. **Methods:** Prospective data collection, level I trauma centre, 1 January 2018–31 December 2019. Inclusion criteria: Primary admission for trauma, injury severity score (ISS) ≥ 16, age ≥ 16. Parameters: Age, ISS, temperature, presence of head injury by the Glasgow Coma Scale (GCS). Outcomes: SIRS and sepsis within 21 days and early death within 72 h after hospitalisation. Statistics: Area under the receiver operating characteristic (ROC) curve for predictive quality, calibration plots for graphical goodness of fit, Brier score for overall performance of WATSON and TRISS. **Results:** Between 2018 and 2019, 107 patients were included (33 female, 74 male; mean age 48.3 ± 19.7; mean temperature 35.9 ± 1.3; median ISS 30, IQR 23–36). The area under the curve (AUC) is 0.77 (95% CI 0.68–0.85) for SIRS and 0.71 (95% CI 0.58–0.83) for sepsis. WATSON and TRISS showed similar AUCs to predict early death (AUC 0.90, 95% CI 0.79–0.99 vs. AUC 0.88, 95% CI 0.77–0.97; *p* = 0.75). The goodness of fit of WATSON (*X*^2^ = 8.19, Hosmer–Lemeshow *p* = 0.42) was superior to that of TRISS (*X*^2^ = 31.93, Hosmer–Lemeshow *p* < 0.05), as was the overall performance based on Brier score (0.06 vs. 0.11 points). **Discussion:** The validation supports previous reports in terms of feasibility of the WATSON Trauma Pathway Explorer and emphasises its relevance to predict SIRS, sepsis, and early death when compared with the TRISS method.

## 1. Introduction

Large patient databases have been successfully used to assess the clinical course in trauma patients [1,2,3]. In line with these changes and medical improvements, the focus regarding complications has moved from early pulmonary changes to later complications that may determine outcomes, such as sepsis.

Our group recently used a big database to develop a new predictive visual analytics tool for polytrauma patients and has presented a proof of concept (IBM WATSON Trauma Pathway Explorer) [1]. It has become evident that an existing database can be used to select certain parameters that help determine certain risk profiles. They reconfirm that a combination of indicators of acute haemorrhage, coagulopathy, acid–base changes, and indirect signs of soft tissue injury remain evident [1]. In the current follow-up study, we used a validation set of patients and compared the prognostic accuracy of the WATSON Explorer with the trauma and injury severity score (TRISS) as an established prediction tool. Moreover, we looked at SIRS and sepsis within 21 days since admission and early death within 72 h since admission using Sankey diagrams [4]. Thereby, a comparison of the predictive value regarding mortality could be made in comparison with TRISS, which represents the closest predictive tool in terms of simplicity. Within WATSON, it is possible to compare the predictive abilities for mortality, sepsis and SIRS.

Therefore, the aim of this study was threefold:(i)to validate the WATSON Explorer in a different patient population;(ii)to expand the predictive capacity from mortality to clinical complications, such as SIRS and sepsis;(iii)to compare aspects of prediction with the TRISS methodology.

## 2. Methods

The study was conducted according to the guidelines for good clinical practice and follows the Helsinki guidelines. The research was based on the TRIPOD Statement, a guideline for multivariable prediction models [5].

The analysis of patient records has been approved by the ethical committee upon the development of the database (Nr. StV: 1-2008) and reapproved to develop the WATSON Trauma Pathway Explorer (BASEC: 2021-00391). For model validations, there is little evidence for calculating sample size. Consecutively, multiple injured patients were prospectively enrolled if they were treated after the development period of the WATSON analytics tool. All patients were recruited during two full years between 1 January 2018, and 31 December 2019, including all seasons, in a single-level I trauma centre.

### 2.1. Inclusion/Exclusion Criteria

Eligibility criteria for the participants were age ≥ 16 years and ISS ≥ 16. Patients admitted primarily and only those with complete datasets were included. Patients referred from another hospital were excluded, and those with missing prediction data required for the calculation of TRISS or WATSON criteria, as well as non-survivors on the scene these patients were also excluded. In the initial care of polytrauma patients, the temperature was not always taken before entering the shock room or in the shock room itself. Since the temperature at admission is mandatory for WATSON’s predictions, these patients were not included in the validation and no imputation method could be considered. Early deceased patients were not excluded for WATSON’s prediction of SIRS or sepsis.

### 2.2. Definitions

The injury severity score (ISS), based on the Abbreviated Injury Scale (AIS; update 2008 version), was used to determine regional injuries and to grade the general severity of trauma [6]. Shock states (I–IV) were defined according to the criteria used by ATLS, an established scoring system widely used for medical teaching. SIRS was defined as the presence of two or more of the following criteria: body temperature > 38 °C or < 36 °C, heart rate > 90 bpm, respiratory rate > 20 breaths/min or PaCO_2_ < 32 mmHg, and white blood cell count >12,000/µL or <4000/µL. The same criteria apply for the presentation of sepsis, with an additional infectious focus [7,8,9]. TRISS includes six physiological and clinical parameters. Estimated survival probability by TRISS was based on the logarithmic regression formulas listed below, according to the TRISS Coefficients 2009 Revision [10]:For blunt trauma = −0.4499 + 0.8085 × RTS − 0.0835 × ISS − 1.7430 × Age Index
For penetrating trauma = −2.5355 + 0.9934 × RTS − 0.0651 × ISS − 1.1360 × Age Index

### 2.3. Clinical Course

Clinical complications focused on SIRS and sepsis and were measured each day from admission for 21 days. All data needed for SIRS and sepsis criteria were documented for each day. If a day displayed more than one value for temperature, heart rate, respiratory rate, or white blood cell count, the clinically most abnormal number was used for the data extraction. All outcomes were ascertained by one investigator. The parameters used for the WATSON Trauma Pathway Explorer and TRISS are compared in Table 1. Four predictors were considered for the validation of the visual analytics tool: age, temperature, ISS, and the AIS filter for head injury. Head injury was included because it is one of the strongest predictive variables [11]. AIS scores for each body region were extracted from the admission report to calculate the ISS score. For the calculation of the revised trauma score (RTS), which is a component of TRISS, the earliest documented values for Glasgow Coma Scale (GCS), systolic blood pressure, and respiratory rate were used. All data were visible in the out-of-hospital rescue service log. All other data needed for the validation were retrieved from the admission report.

### 2.4. Statistics

The baseline characteristics of the patients’ sample were described through means with standard deviation (SD) for numerical variables, medians with interquartile ranges (IQR) for ordinal data, and percentages for binary variables.

For descriptive statistics, the patient sample was also divided into two subgroups: survivors versus early deceased within 21 days after admission. Unpaired *t*-test for numerical variables and Mood’s median test for ordinal variables assessed the differences between these groups.

We assessed the predictive performance of the WATSON-based visual analytics tool and TRISS by examining measures of discrimination and calibration.

Discrimination refers to how strongly a predictive model can distinguish from current patient data whether a patient will or will not experience a given outcome [11]. The receiver operating characteristic (ROC) curve was calculated [12]. We considered an AUC of at least 0.7 to be reasonable discrimination [13,14,15]. The statistical precision of the measurements is quantified by confidence intervals (CI), which have been determined by bootstrapping with 2000 repetitions. As TRISS presents estimated survival rates, for the comparison with WATSON, all values were subtracted from 1 to provide predicted death rates. To quantify the difference between the two AUCs, a *p*-value was calculated. When using the predefined significance level of 5%, a *p*-value < 0.05 indicates that the AUCs are not equal [16]. A cut-off point of 0.5 was used to classify predicted probabilities as events or non-events.

Calibration describes whether the predictive model can discriminate over the entire range of outcome probabilities [11]. The plotting of the predicted outcome probabilities on the *x*-axis versus the observed outcomes on the *y*-axis makes it possible to graphically assess the calibration of the prediction models [12]. A perfect prediction lies on the 45-degree line [12]. The Hosmer–Lemeshow test was used to quantify the calibration of the models, following a chi-square distribution. The range of predicted early death was divided into 10 groups. The smaller the chi-square value and the higher the *p*-value, the better the fit. A chi-square value of zero indicates perfect calibration. [15].

For the overall model’s performance, we used the Brier score, which summarises the discrimination and calibration by defining the mean squared error of prediction [17]. The score ranges from 0 to 1, where 0 indicates perfect accuracy. Data were analysed using Python version 3.6.9 (Python Software Foundation, Wilmington, DE, USA).

### 2.5. Development of the Model and Validation

The general characteristics of the validation group were tabularly compared with the patients used for the development of the tool (Table 2). As the WATSON-based visual analytics tool for polytrauma patients is based on a local trauma data bank, we assumed that all patients included in the prediction validation were treated according to the same clinical guidelines. The eligibility criteria were the same for the patients used in the development and validation of the prediction model. The definition of the outcomes in the validation sample was the same as the original outcome definition in the development group. The architectural components of the WATSON Trauma Pathway Explorer are shown in Figure 1 [1].

## 3. Results

### 3.1. General Characteristics

The outcome validation encompassed 107 polytrauma patients, 33 female and 74 male (Table 2). The age distribution ranged between 16 years and 89 years (mean 48.3 ± 19.7). The gender distribution followed the epidemiology of polytrauma in Europe [18]. In most cases, the trauma mechanism was blunt (99.1%). Road traffic accidents accounted for the majority, followed by falls. 75 participants suffered additional head injury. The range of the ISS score in the 107 polytrauma patients varied from ISS 17 to ISS 75 (median 30, IQR 23–36), while the body temperature at admission ranged from 28.4 °C to 37.4 °C (mean 35.9 ± 1.3). No patient was overheated > 38 °C. All four ATLS shock classes were present in the 107 patients (median 1, IQR 1–3).

Patients that deceased early showed a similar age as survivors (*p* = 0.063). ISS, temperature, GCS, systolic blood pressure, and respiratory rate showed significantly worse results in non-survivors (*p* < 0.05). Based on the GCS, systolic blood pressure, and respiratory rate, the RTS ranged from 0 to 7.84 (median 6.90, IQR 5.97–7.84).

The general demographics of the 107 polytrauma patients correspond with the patients incorporated into the database used by the WATSON Trauma Pathway Explorer. The demographic data of the validation and development group are described in Table 2 [1].

### 3.2. Model Performance

The predictive performance of the WATSON Trauma Pathway Explorer and TRISS is summarised in Table 3.

SIRS criteria were assessed on each of the 21 days since admission. As described above, at least two of four clinical findings are needed for the definition of SIRS. In total, 82 patients developed SIRS, while 25 patients did not meet any SIRS criteria. SIRS was present at admission in 55 cases. In some patients, SIRS occurred after several days. Only four patients suffered SIRS each of the 21 days. The 107 predictions by WATSON for this outcome ranged from 33% to 100%. All patients with a predicted probability of more than 90% for SIRS developed the clinical manifestation. As is apparent in the ROC curve, the AUC is near 0.8 (Figure 2).

Sepsis developed in 13 cases. In those patients, SIRS was present for at least five days. WATSON’s range for predicted sepsis in the analysed patient group was between 0% and 100%. The predictive accuracy was lower for sepsis than for SIRS or early death. The AUC amounts to approximately 0.7 (Figure 3).

The 11 patients who died within 72 h had mostly experienced a polytrauma with head injury. Early death and sepsis did not occur together in this prediction validation. Both WATSON and TRISS showed good discrimination for the prediction of early death with an AUC of >0.8 (Figure 4). The predictive quality of the WATSON Trauma Pathway Explorer was slightly higher than the reference score (AUC 0.90, 95% CI 0.79–0.99 vs. AUC of 0.88, 95% CI 0.77–0.97; Table 3). However, the difference between the two AUCs was not significant (*p* = 0.75). The WATSON Trauma Pathway Explorer’s predictions for early death in the 107 analysed patients ranged from 0% to 100%, and the death rates estimated by TRISS ranged from 1.13% to 99.94%. Graphical illustration of early death may suggest a similar calibration of the WATSON Trauma Pathway Explorer and TRISS (Figure 4). However, the Hosmer–Lemeshow test presented better goodness of fit for the WATSON Trauma Pathway Explorer than for TRISS (*X*^2^ = 8.19, Hosmer–Lemeshow *p* = 0.42 vs. *X*^2^ = 31.93, Hosmer–Lemeshow *p* < 0.05; Table 3). WATSON’s calibration was particularly stronger at higher death probabilities. The Brier score of WATSON was better than that of TRISS in predicting early death (0.06 vs. 0.11 points).

## 4. Discussion

Worldwide, trauma surgeons assert that the primary goal of trauma care to improve survival rates. They often focus on reducing the number of early deaths, as these are considered to be preventable [19,20]. For this reason, several trauma scoring systems have been developed to estimate survival rate in trauma patients, including the Trauma and Injury Severity Score (TRISS) or A Severity Characterization of Trauma (ASCOT) [21]. TRISS is a scoring system used for the determination of survival probability in blunt and penetrating trauma and is one of the most commonly used scores in this area [22,23]. It combines physiological and anatomical trauma scoring systems. The score comprises the revised trauma score (RTS) for the assessment of the physiological state of a polytrauma patient and the injury severity score (ISS) for the anatomical severity of the injuries. RTS includes the GCS, systolic blood pressure, and respiratory rate. Furthermore, TRISS considers the type of trauma, i.e., penetrating versus blunt [24]. The coefficients in TRISS are estimated by logistic regression and have been revised several times [10,23,25]. TRISS has shown accurate survival estimation for trauma patients in several studies with a discriminative AUC > 0.8 [26,27,28,29,30]. However, a recent publication demonstrated the benefits and outperformance of machine learning for predicting outcomes in trauma patients compared to established trauma scoring systems [31,32,33,34].

For these reasons, we had decided to use TRISS for comparison. The new WATSON application for the assessment of polytrauma patients demonstrates that a tool is capable of predicting different outcomes of patients who have sustained multiple injuries.

Our main results regarding the validation of the predicted outcomes are as follows:

The prediction of the WATSON-based visual analytics tool for early death corresponded to the effective clinical outcome in approximately 90% of the analysed polytrauma patients, which was similar to the discriminative performance of TRISS. The WATSON Trauma Pathway Explorer, however, was better calibrated to the test data.

Within WATSON’s prediction options, its validity for early death was better than for SIRS and sepsis (80% and 70%, respectively). Furthermore, the graphical calibration for sepsis suggested lower goodness of fit than for SIRS and early death.

Several scores exist, particularly for the survival probability after trauma, such as TRISS, which has been compared with the predicted outcome of the WATSON Trauma Pathway Explorer in this study. There are few and inconsistent scores for the prediction of sepsis in trauma patients. The traumatic sepsis score (TSS) was developed to predict sepsis risk following trauma, with an AUC of 0.79. The score included the ISS, GCS, temperature, heart rate, albumin, international normalized ratio (INR), and C-reactive protein (CRP) [35]. Other scores and parameters, such as the new injury severity score (NISS), the Acute Physiology and Chronic Health Evaluation Score II (APACHE II), and the prothrombin time, were evaluated regarding the predictive ability for sepsis in polytrauma patients, showing AUCs of 0.77, 0.82, and 0.74, respectively [36].

An advantage of our prediction model is the implementation of the existence or absence of a head injury, which is the number one killer [11]. On the other hand, it lacks physiological parameters such as systolic blood pressure or respiratory rate, both of which are included in TRISS. One way to improve early death estimation in our model could be the addition of blood parameters such as lactate and pH to the existing predictive parameters. Another way to improve predictive performance may be the inclusion of out-of-hospital traumatic cardiac arrest. Prehospital cardiac arrest in trauma patients is associated with poor effects on patient outcomes [37,38]. Previous studies have stated the importance of high-sensitivity troponin T (hs-TnT), red cell distribution width (RDW) or C-reactive protein (CRP) as predictors of a cardiogenic shock or multiple organ dysfunction syndrome [39]. These parameters, especially CRP, may also play an important role in predicting SIRS or sepsis, and could be examined in the future regarding predictive quality.

Several attempts to improve or outperform TRISS have shown mixed results. In a study by Domingues et al., the original TRISS equation was compared with three new adjustments to TRISS, which included best motor response, peripheral oxygen saturation, or new injury severity score. The newly proposed TRISS adjustments showed no difference in predictive performance (*p* > 0.5) [40]. Becalick et al. compared TRISS with artificial intelligence techniques. However, the results showed significantly better discriminative values for TRISS [11].

Only a small percentage of all physicians assume that they correctly assess the prediction of traumatic head injuries [41]. Nonetheless, an accurate prediction of trauma patients regarding early death is essential for treatment decisions and other implications [33]. Studies dealing with machine learning in traumatology share similar features for model development to the WATSON-based visual analytics tool for polytrauma patients. These include age, temperature, ISS, respiratory rate, or heart rate [31].

Our study has several limitations. The determination of the AIS and the resulting ISS has certain interobserver weaknesses, which may play a role in our assessment. However, since all AIS scores have been determined by one investigator, this limits the subjective variance between the two models compared.

No blinding experiment was conducted. However, the outcome early death does not leave any room for interpretation and is clearly defined in time.

The study only considered death within an arbitrary 72 h since admission. In most cases, a trauma-related death in a hospital occurs in the first hours or first few days after admission. Nevertheless, some patients in this validation study sustained death after 72 h due to trauma-related complications and were therefore not recorded.

The high Hosmer–Lemeshow *p*-value for the WATSON Trauma Pathway Explorer may be affected by the test having lower power to detect misspecification since our sample size is rather small.

Finally, it must be emphasised that this study represents an internal validation with a limited sample size in the same institution and with the same practice patterns as in the development of the WATSON Trauma Pathway Explorer. We feel that the results may therefore not translate into other healthcare systems until there is accurate external validation.

In summary, the WATSON Trauma Pathway Explorer has three applications. Firstly, WATSON strives to be an educational tool, helping to show young residents the correlation between predictive values and outcomes, in particular, early death, SIRS, and sepsis. Secondly, the visual analytics tool can trigger research, because it has the potential to reveal clinical relations or observations that have not been understood so far. Furthermore, it brings new insights into old parameters and might lead to new interpretations [1]. For example, in research into cancer pathway signalling, WATSON suggested connections that would otherwise not have been considered [42]. Finally, the WATSON Trauma Pathway Explorer could act as a supporting tool in clinical decision-making. However, the visual analytics tool is not meant to be a piece of advice, and no clinical recommendation about the current patient is made.

Our findings show how big data-based systems have the potential to improve or replace established scores and to give us a deeper understanding of clinical relations in traumatology. This visual analytics tool, a variant of AI, will provide the foundation for personalized medicine in polytrauma patients.

## Figures and Tables

**Figure 1 jcm-10-02115-f001:**
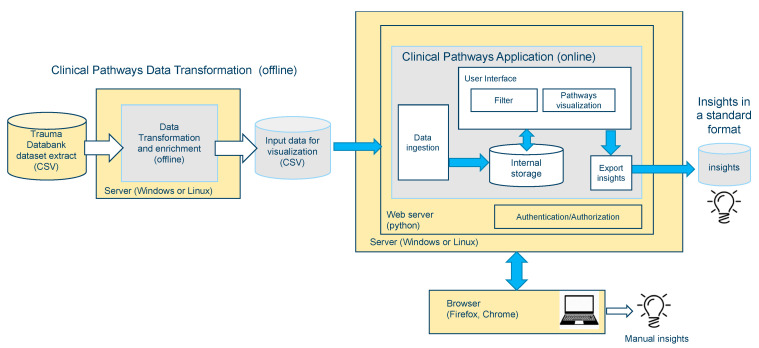
The digital architecture of the Sankey visual analytics tool modified for use in polytraumatized patients.

**Figure 2 jcm-10-02115-f002:**
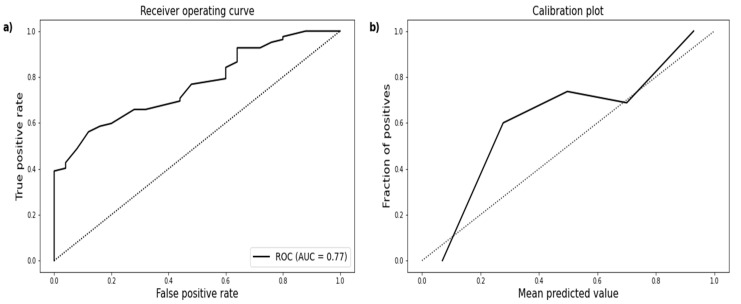
(**a**) ROC curve demonstrating an AUC of 0.77 (95% CI 0.68–0.85) for predicted SIRS. (**b**) Calibration plot showing the distribution of the observed SIRS and the corresponding prediction.

**Figure 3 jcm-10-02115-f003:**
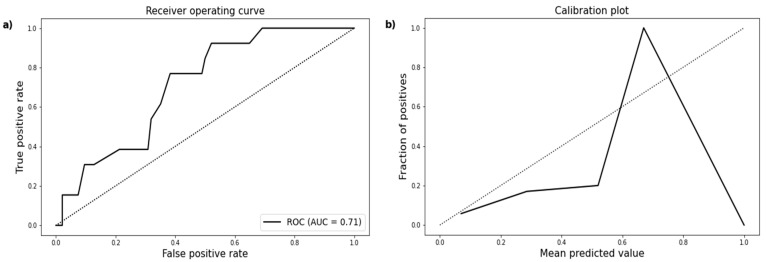
(**a**) ROC curve demonstrating an AUC of 0.71 (95% CI 0.58–0.83) for predicted sepsis. (**b**) Calibration plot showing the distribution of observed sepsis and the corresponding prediction.

**Figure 4 jcm-10-02115-f004:**
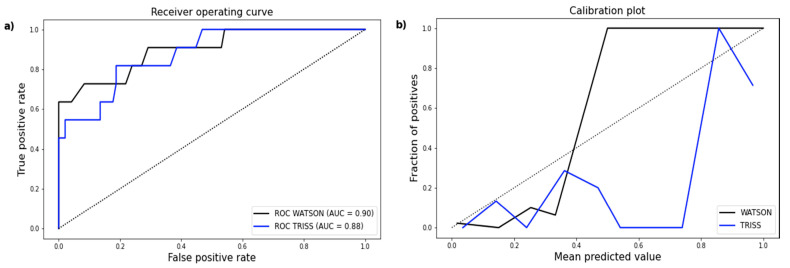
(**a**) ROC curve demonstrating an AUC of 0.90 (95% CI 0.79–99) for early death predicted by WATSON and an AUC of 0.88 (95% CI 0.77–0.97) for early death predicted by TRISS. (**b**) Calibration plot showing the distribution of observed early death and the corresponding prediction by WATSON and TRISS.

**Table 1 jcm-10-02115-t001:** Predictors used in WATSON and TRISS.

Predictors in WATSON	Predictors in TRISS	Measurement
Age	Age	Admission report, numerical data
Temperature	—	Admission report, numerical data
ISS	ISS	Admission report, ordinal data
AIS filter for head injury	—	Admission report, binary data
—	GCS	Rescue service log, ordinal data
—	Systolic blood pressure	Rescue service log, numerical data
—	Respiratory rate	Rescue service log, numerical data
—	Type of trauma	Admission report, binary data

TRISS = trauma and injury severity score; ISS = injury severity score; AIS = Abbreviated Injury Scale; GCS = Glasgow Coma Scale.

**Table 2 jcm-10-02115-t002:** General characteristics of the validation group and the development group.

	Validation Group				Development Group
	Patient Sample *n* = 107	Survivors *n* = 96	Non-Survivors *n* = 11	*p*-Value	Patient Sample *n* = 3647
Age (mean, SD)	48.3 ± 19.7	47.1 ± 19.0	58.7 ± 24.0	0.063	45.8 ± 20.2
Male	69.2% (*n* = 74)	69.8 % (*n* = 67)	63.6% (*n* = 7)	—	73.5 % (*n* = 2680)
Blunt trauma	99.1% (*n* = 106)	100% (*n* = 96)	90.9% (*n* = 10)	—	91.3% (*n* = 3329)
ATLS shock class (median, IQR)	1 (1–3)	1 (1–3)	1 (1–3.5)	0.149	1 (1–2)
ISS (median, IQR)	30 (23–36)	29 (22–34.5)	42 (31–66)	0.009	25 (17–34)
Temperature at admission (mean, SD)	35.9 ± 1.3	36.0 ± 1.2	34.9 ± 1.6	0.007	35.5 ± 1.7
Head injury	70.1% (*n* = 75)	67.7% (*n* = 65)	90.9% (*n* = 10)	—	76.2% (*n* = 2780)
SIRS (within 21 days)	76.6% (*n*= 82)	75.0% (*n* = 72)	90.9% (*n* = 10)	—	83.5% (*n* = 3044)
Sepsis (within 21 days)	12.1% (*n* = 13)	13.5% (*n* = 13)	0% (*n* = 0)	—	15.0% (*n* = 546)
Early Death (within 72 h)	10.3% (*n* = 11)	—	—	—	19.4% (*n* = 709)
GCS at patient contact (median, IQR)	13 (8.5–15)	14 (9–15)	3 (3–9.5)	<0.001	—
SBP at patient contact (mean, SD)	119 ± 37	122 ± 32	98 ± 66	0.039	—
RR at patient contact (mean, SD)	17.3 ± 6.7	17.8 ± 6.5	12.9 ± 7.4	0.022	—
RTS at patient contact (median, IQR)	6.90 (5.97–7.84)	7.11 (6.38–7.84)	4.09 (3.73–5.71)	<0.001	—

SD = standard deviation; IQR = interquartile range; ATLS = Advanced Trauma Life Support; ISS = injury severity score; GCS = Glasgow Coma Scale; SBP = systolic blood pressure; RR = respiratory rate; RTS = revised trauma score.

**Table 3 jcm-10-02115-t003:** Results of the ROC analysis for WATSON versus TRISS.

	AUC	H-L Statistics	Brier SCORE
SIRS by WATSON	0.77 (95% CI 0.68–0.85)	*X*^2^ = 5.24, *p* = 0.73	0.15
Sepsis by WATSON	0.71 (95% CI 0.58–0.83)	*X*^2^ = 12.14, *p* = 0.14	0.12
Early Death by WATSON	0.90 (95% CI 0.79–0.99)	*X*^2^ = 8.19, *p* = 0.42	0.06
Early Death by TRISS	0.88 (95% CI 0.77–0.97)	*X*^2^ = 31.93, *p* < 0.05	0.11

AUC = area under curve; H-L = Hosmer–Lemeshow; CI = confidence intervals; TRISS = trauma and injury severity score.

## Data Availability

All data are available upon reasonable request. None of these data are available for broad public. All data are stored in in the KISIM System (Clinical information system) of the University Hospital of Zurich.

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
