# Peer review of "Validation of a Visual-Based Analytics Tool for Outcome Prediction in Polytrauma Patients (WATSON Trauma Pathway Explorer) and Comparison with the Predictive Values of TRISS"

_jcm, 2021, doi:10.3390/jcm10102115_

Round 1

Reviewer 1 Report

Succinctly written article with sound statistical methods. 

Patients with missing prediction data were excluded. Did you consider multiple imputation or any other method instead of just excluding the patients with missing data? Could you give any information regarding the excluded patients?

How did you determined study group size for the validation cohort?

Were early deceased patients excluded for WATSON prediction of sepsis?

Author Response

Dear Sir, thank You very much for reviewing our manuscript.

  1. An imputation would have changed the data set towards virtual average values. This is the reason why we left them. The 2.0 version of this application under vent a data enrichment and will have much more patients included. The included patients had missing data in the required time and parameter characteristics. I.e. at the admission to the trauma bay. These were not very severely injured patients. Just handling system mistakes.
  2. The sample size was set arbitrarily, high enough to represent the population of the databank Watson is running on.
  3. The early deceased patients (<72h) if they did not suffer sepsis were not included into the cohort of septicians.
  4. All changes were incorporated into the manuscript.

Reviewer 2 Report

Thank you for the opportunity to review your manuscript entitled " Validation of a visual-based analytics tool for outcome predic-2 tion in polytrauma patients (WATSON Trauma Pathway Ex-3 plorer) and comparison with the predictive values of TRISS".

Previous studies have indicated that high-sensitivity troponin T (hs-TnT),  red cell distribution width (RDW) or C-reactive protein (CRP) are predictors of a cardiogenic shock or multiple organ dysfunction syndrome (MODS) (1-3). The knowledge of the predictors of cardiogenic shock or MODS are extremely important, because it enables the identification of patients at risk of this complications and the early implementation of treatment, which increases the patient's chances of survival.

Abstract, title and references.

The aim of the study is clear. The title is informative and relevant. The references are relevant, recent, and referenced correctly.

Please complete your references with the following article:

1.doi: 10.1097/SHK.0000000000001360

Introduction

It is clear what is already known about this topic. The research question is clearly outlined.

Methods

The process of subject selection is clear. The variables are defined and measured appropriately. The study methods are valid and reliable. There is enough detail in order to replicate the study.

Discussion and Results

The results are discussed from multiple angles and placed into context without being overinterpreted. The conclusions answer the aims of the study. The conclusions supported by references and results. The limitations of the study are opportunities to inform future research.

Overall. The study design was appropriate to answer the aim.

The manuscript is well written and a stimulus for the readership.

Minor revisions:

Were the parameters of hs-Troponin T, RDW or CRP assessed in the study group?

* Please add the following reference:

1.doi: 10.1097/SHK.0000000000001360

Author Response

Dear Sir thank You very much for Your effort to sight our article.

  1. The refference was added
  2. The hs-Troponin, RDW and CRP were in this validation study not adressed. However, they are accesable in this Watson application. We are actually statistically analysing new interpretation (predictive) abilities of CRP in polytrauma patients.

With best regards

Ladi Mica